# Hydrolyzed Flavonoids from *Cyrtosperma johnstonii* with Superior Antioxidant, Antiproliferative, and Anti-Inflammatory Potential for Cancer Prevention

**DOI:** 10.3390/molecules27103226

**Published:** 2022-05-18

**Authors:** Ornchuma Naksuriya, Krai Daowtak, Singkome Tima, Siriporn Okonogi, Monika Mueller, Stefan Toegel, Ruttiros Khonkarn

**Affiliations:** 1Department of Pharmaceutical Science, Faculty of Pharmacy, Chiang Mai University, Chiang Mai 50200, Thailand; ornchuma.n@cmu.ac.th (O.N.); siriporn.okonogi@cmu.ac.th (S.O.); 2Cellular and Molecular Immunology Research Unit, Faculty of Allied Health Sciences, Naresuan University, Phitsanulok 65000, Thailand; kraid@nu.ac.th; 3Department of Medical Technology, Faculty of Associated Medical Sciences, Chiang Mai University, Chiang Mai 50200, Thailand; singkome.tima@cmu.ac.th; 4Department of Pharmaceutical Technology and Biopharmaceutics, University of Vienna, Althanstrasse 14, 1090 Vienna, Austria; monimueller@gmx.at; 5Karl Chiari Lab for Orthopaedic Biology, Department of Orthopedics and Trauma Surgery, Medical University of Vienna, Waehringer Guertel 18–20, 1090 Vienna, Austria; stefan.toegel@meduniwien.ac.at; 6Research Center of Producing and Development of Products and Innovations for Animal Health and Production, Chiang Mai University, Chiang Mai 50200, Thailand

**Keywords:** *Cyrtosperma johnstonii*, quercetin, isorhamnetin, flavonoid glycosides, antioxidant, cytotoxicity, anti-inflammatory, synergism

## Abstract

*Cyrtosperma johnstonii* is one of the most interesting traditional medicines for cancer treatment. This study aimed to compare and combine the biological activities related to cancer prevention of the flavonoid glycosides rutin (RT) and isorhamnetin-3-o-rutinoside (IRR) and their hydrolysis products quercetin (QT) and isorhamnetin (IR) from *C.*
*johnstonii* extract. ABTS and MTT assays were used to determine antioxidant activity and cytotoxicity against various cancer cells, as well as normal cells. Anti-inflammatory activities were measured by ELISA. The results showed that the antioxidant activities of the compounds decreased in the order of QT > IR > RT > IRR, while most leukemia cell lines were sensitive to QT and IR with low toxicity towards PBMCs. The reduction of IL-6 and IL-10 secretion by QT and IR was higher than that induced by RT and IRR. The combination of hydrolysis products (QT and IR) possessed a strong synergism in antioxidant, antiproliferative and anti-inflammatory effects, whereas the combination of flavonoid glycosides and their hydrolysis products revealed antagonism. These results suggest that the potential of the combination of hydrolyzed flavonoids from *C. johnstonii* can be considered as natural compounds for the prevention of cancer.

## 1. Introduction

Cancer is the first or second leading cause of death in patients under the age of 70 in 112 of 183 countries, including Thailand, and the cause of almost 10 million deaths globally in 2020 [1]. Breast, cervical, colorectal, liver, and lung cancers account for more than half the cases of cancer in Thailand. The age standardized incidence rate (ASR) projected that breast and lung cancer cases will increase by approximately 50% and 25%, respectively, from the year 2012 to 2025 [2]. A stressful lifestyle, eating habits, smoking, an environment with increasing levels of pollution, radiation, and pesticides are the major causes of cancer [3]. Cancer is characterized by the accumulation and uncontrolled proliferation of abnormal cells. It is hypothesized that carcinogenesis has three stages: the initiation stage begins with cells being stimulated by various carcinogens including oxidative stress, which causes DNA damage within cells; the promotion stage involves the expansion of modified DNA within cells; and the progression stage is characterized by the transformation of modified DNA within cells into uncontrolled tumor cell growth, with additional genetic alterations [4]. Treatments for cancer have usually included chemotherapy, radiation therapy, stem cell transplant, and surgery. Nevertheless, chemotherapy is often restricted by dose-limiting toxicity and severe side effects such as cardiotoxicity, hepatotoxicity, nephrotoxicity, neurotoxicity, and myelosuppression [5,6]. Many patients fail to respond to the current chemotherapy. Some of them show incomplete response or relapse that is caused by drug resistance after the treatment [7]. The search for alternative therapies has become an important topic in biomedical research. The excessive production of reactive oxygen species (ROS) is known as one of the leading causes of cancer. The changing in redox balance and redox signaling contributes to cancer progression and resistance to treatment [8]. Antioxidants from bio-resources, especially flavonoids, can reduce oxidative stress directly by ROS scavenging and metal ion chelating, leading to the prevention of DNA damage [9]. Flavonoids also activate antioxidant enzymes (such as glutathione, superoxide dismutase, and catalase) and suppress pro-oxidant enzymes (such as NADPH oxidase, lipoxygenase, and xanthine oxidase) as indirect effects [10]. Moreover, flavonoids can play a role as pro-oxidants in cancer cells, which may prevent the progression of cancer through mechanisms including cell proliferation, apoptosis induction, cell cycle arrest, and reversal of the multidrug resistance phenomenon. Besides, flavonoids are also related to the prevention of chronic inflammation that can foster cancer development, possibly due to the suppression of pro-inflammatory cytokines, including tumor necrosis factor (TNF), interferon (INF)-γ, interleukins (such as IL-1, IL-6, IL-12 and IL-18), and anti-inflammatory cytokines (such as IL-4, IL-10, IL-13, IFN-α) [11]. Investigation of new substances, especially from natural resources such as medicinal plants, has been attractive due to their potential anticancer activities, their reduced ability to evoke drug resistance, and their lack of serious side effects on normal cells [12]. *Cyrtosperma johnstonii* (*C. johnstonii*) is a plant that is found in many countries of southeast Asia, including Thailand. The rhizome of this plant has been used in Thai traditional medicine as an appetite stimulant, blood tonic, and cancer therapeutic agent. Our previous study investigated the rhizome extracts of *C. johnstonii* in several in vitro models of antioxidant and cytotoxic activities and found respective activities that were mainly due to its flavonoid glycoside components [13]. By the way, several previous studies demonstrated that the hydrolysis product of flavonoid glycoside, the so-called aglycone, also exhibits antioxidant and cytotoxic activities [14,15]. It is hypothesized that the combination of the hydrolysis product of flavonoid glycosides extracted from *C. johnstonii* rhizomes might exert a higher benefit toward biological actions. However, to the best of our knowledge, so far there are no studies on the biological effect of flavonoids from *C. johnstonii* rhizomes in combination with other flavonoids and their hydrolysis products. To bridge this gap in knowledge and evaluate its potential value in cancer prevention, flavonoid glycosides and their hydrolysis products were compared in terms of their antioxidant, antiproliferative and anti-inflammatory activities. The combination effect was also evaluated in terms of its potential cancer prevention effects. The individual and combined effects of flavonoid glycosides and their hydrolysis products on antioxidant activity were determined by a radical scavenging assay. For antiproliferative tests, breast, cervical, and leukemic cancer cell lines were used as cancer cell models and the safety profiles in normal cells were investigated. Finally, the present study investigated anti-inflammatory activity by the reduction of pro- and inflammatory cytokine (IL-6 and IL-10) in macrophage cells.

## 2. Materials and Methods

### 2.1. Materials

Dried powder of *C. johnstonii* rhizomes was collected from the central part of Thailand. Quercetin (QT) and isorhamnetin (IR) were purchased from Sigma-Aldrich (St. Louis, MO, USA). Ortho-phosphoric acid, hexane, ethyl acetate, acetone, methanol, acetonitrile, trifluoroacetic acid, potassium persulfate, ethanol, and dimethylsulfoxide (DMSO) were obtained from Merck (Darmstadt, Germany). 2,2′-azino-bis3-ethylbenzothiazoline-6-sulfonic acid) diammonium salt (ABTS), 2,4,6-tris(2-pyridyl)-s-triazine (TPTZ), Trolox, 3-(4,5-dimethylthiazol-2-yl)-2,5-diphenyltetrazolium bromide (MTT) were purchased from Sigma-Aldrich (St. Louis, MO, USA). RPMI-1640, Dulbecco’s modified eagle medium (DMEM), trypan blue and penicillin/streptomycin were obtained from Invitrogen^TM^ (Grand Island, NY, USA). Fetal bovine serum was purchased from GIBCO-BRL (Grand Island, NY, USA).

### 2.2. Standardization of Sample Plant Extracts by HPLC Analysis

Dried powder of *C. johnstonii* rhizomes was extracted by a fraction of hexane, ethyl acetate, acetone, and methanol [13]. The acetone fraction was selected to separate rutin (RT) and isorhamnetin-3-o-rutinoside (IRR) by preparative TLC. Next, 100 mg of plant extracts were dissolved in 1 mL of extracted solvent for HPLC analysis. The solution was filtered through a 0.45 µm filter. The sample was analyzed with a Shimadzu CL-20AD HPLC system with a SPD-M20A photodiode array detector. A C18 column (250 × 4.6 mm, 5 µm particle size) with a guard column was used. The elution of the constituents, including a gradient of two solvents denoted as A and B, was conducted. A was acetonitrile, whereas B was 0.1% *v*/*v* aqueous ortho-phosphoric acid. The gradient program was as follows: 100% B to 90% B in 5 min, 90% to 75% B in 5 min, constant at 75% B for 20 min, 75% B to 50% B in 18 min, constant at 50% B for 2 min, and 50% B to 100% B in 5 min. There was 15 min of post-run for reconditioning. The flow rate used was 1.0 mL/min at room temperature and the injection volume was 20 µL. The retention time and UV spectrum of major peaks were analyzed. The flavonoid glycosides were kept in vacuum desiccators overnight to remove residual solvent. Hydrolysis products of flavonoid glycosides were prepared by chemical hydrolysis using 0.2 M and 2.0 M of trifluoroacetic acid (TFA) at 100 °C for 30 min to obtain QT and IR (Figure 1). The hydrolysis products of flavonoid glycosides were characterized by HPLC with the same instrument and column. Briefly, 0.1 mg of hydrolysis products of flavonoid glycosides samples or standard QT or IR were dissolved in 1 mL of methanol. The solution was then filtered through a 0.45 µm filter. The gradient elution of the constituents was conducted as elution A and elution B. The elution A was acetonitrile whereas the elution B was 0.1% *v*/*v* aqueous ortho-phosphoric acid and 5% *v*/*v* of acetonitrile. The gradient program was as follows: 0% A to 15% A in 2 min, 15% to 40% A in 15 min, 40% to 60% A in 3 min, 60% to 100% A in 1 min, 100% to 0% A in 1 min and constant at 0% A for 2 min. The column temperature was 40 °C. The flow rate used was 0.50 mL/min and the injection volume was 20 µL. 

### 2.3. Determination of the Antioxidant Activity of Individual Compounds

The antioxidant activity based on the ROS scavenging activity of compounds was performed based on the decolorization of ABTS radical cation as described by Okonogi et al. (2013) [13]. ABTS free radicals were generated by the reaction of ABTS and potassium persulfate in the solution. Briefly, 20 µL of ethanolic solution of QT (12.5–200 µM), RT (200–1000 µM), IR (100–500 µM), and IRR (500–3000 µM) was added to the microtiter plate. Next, 180 µL of ABTS solution was then added. Trolox was used as positive control while ethanol was used as a negative control. The mixtures were kept for 5 min. After that, the absorbance was measured by a microtiter plate reader (Bio-Rad model 680, Hercules, CA, USA) at 750 nm. The antioxidant activity was measured as the 50% effective concentration (EC_50_) The antioxidant activity of the individual compound was calculated by the following equation (Equation (1)): % Antioxidant activity = [(Abs_control_ − Abs_sample_)/Abs_control_] × 100(1)
where Abs_control_ is the absorbance value of ABTS solution and Abs_sample_ is the absorbance value of the test sample with ABTS solution. The antioxidant potential was expressed as Trolox equivalent antioxidant capacity (TEAC) in mM of a Trolox solution whose antioxidant capacity is equivalent to 1.0 mg of the compound.

### 2.4. Determination of the Antioxidant Activity of Combined Compounds 

For the combination samples, the concentrations of the main sample were varied while the concentration of the additional sample was fixed at 20% effective concentration (EC_20_) as described by Naksuriya et al. (2015) [16]. The ABTS method was performed, and the antioxidant activity of the single compound and mixture was calculated by Equation (1). The combination index (CI) was calculated to evaluate the interaction among combination samples as the following equation (Equation (2)):CI = (Mca/Sca) + MCb/SCb)(2)

Mca and MCb are the concentrations of sample A and sample B were used in combination to achieve 50% antioxidant activity. Sca and SCb are the concentrations for single agents to achieve the same effect. A combination index of less than, equal to, and more than 1 indicates synergy, additivity, and antagonism, respectively.

### 2.5. Cytotoxicity Study against Cancer Cells

Human cervical carcinoma (KB-3-1), eosinophilic leukemia (EoL-1), myelomonocytic leukemia (MV4-11), human lymphoblastic leukemia (Molt4), human monocytic leukemia (U937), and human breast cancer (MCF-7) were used as test cancer cells using the MTT assay with some modification of Alley et al. (1988) [17,18,19]. Briefly, the cells were suspended in 100 µL of medium and were plated into the wells of 96-well plates at a density of 1 × 10^5^ cells/well for KB-3-1, Molt-4, U937, and MCF-7, 2 × 10^5^ cells/well for MV4-11 and 5 × 10^5^ cells/well for EoL-1. After 24 h incubation, various concentrations (5–300 μM) of RT, QT, IR, and IRR were added and incubated for 48 h. The supernatant (100 μL) was removed and 15 μL of 5 mg/mL MTT dye in PBS was added to each well and incubated for 4 h. After 4 h, the supernatant was removed, and 200 μL of DMSO was added to each well. The samples were then mixed thoroughly to dissolve the dye crystals. The absorbance at 540 nm was measured with a microtiter plate reader. The percentage of cell viability for each tested sample was calculated using the following equation.
Cell viability (%) = Abs_sample_/Abs_control_ × 100(3)
where Abs_control_ is the absorbance value of the control well and Abs_sample_ is the absorbance value of the test sample well. 

Human immortalized myelogenous leukemia cell line (K562), the corresponding drug-resistant cells with P-glycoprotein (P-gp) overexpression (K562/ADR), were used as a test cancer cell model [13]. The cells were suspended in a 1 mL medium and were seeded into the 24-well plates at a density of 1 × 10^5^ cells/well. After 24 h incubation, solutions with various concentrations of tested samples (3–200 µM), were added and the cells were cultured for a further 72 h. Next, the number of cells was counted by flow cytometry (Coulter^®^ Epics^®^ XLTM). The 50% inhibitory concentration (IC_50_) was evaluated from the dose-response curves of percentages of cell growth inhibition versus the concentration of the test samples. To determine the effects of mixtures, 5 μM of IR was mixed with QT at a concentration of 3–200 μM and 10 μM of QT was mixed with IR at concentration of 3–200 μM. The combination index (CI) was calculated as previously described in the section on the determination of the antioxidant activity of a combination using 50% cell inhibition instead of 50% antioxidant activity.

### 2.6. Cytotoxicity against Normal Cells

Peripheral blood mononuclear cells (PBMC) were collected from the associated medical science (AMS) clinical service center, Chiang Mai University (study code no. AMSEC-63EM-021). The cytotoxicity of the RT, QT, IR, and IRR towards normal PBMC was tested using MTT assay, as previously described in a section cytotoxicity study against cancer cells, with the concentration of 1 × 10^5^ cells/well [17,18,19]. The percentage of cell viability was calculated as previously described in the section on cytotoxicity study against cancer cells. 

### 2.7. Determination of Pro- and Anti-Inflammatory Activity by the Enzyme-Linked Immunosorbent (ELISA) Assay

LPS-stimulated RAW 264.7 cells were used to examine the pro- and anti-inflammatory activity of RT, QT, IR, and IRR. The cell culture was performed along the previous report with slight modification [20]. Briefly, RAW 264.7 cells (American Type Culture Collection, ATCC-TIB-71) were seeded at a density of 5 × 10^5^ cells/well in 12-well plates and incubated for 24 h at 37 °C, 5% CO_2_, and 90% humidity. After that, test samples including RT, QT, IR, and IRR in <0.1% DMSO solution in a medium with a concentration of 100 μM were added, and cells were incubated in the same conditions for a further 3 h before LPS was added at a final concentration of 1 μg/mL. The cells were then incubated in the same conditions for a further 24 h. On the third day, the media was removed and centrifuged at 1500× *g* to remove cells, and the supernatant was aliquoted and stored at 20 °C prior to analysis by ELISA. The concentrations of IL-6, and IL-10, in 100 μL of each cell supernatant were determined by ELISA assay according to the manufacturer’s protocol (R&D Systems) and by Mueller et al. (2015) [21]. Samples were added to a coated plate and incubated for 1 h. After that, the plate was washed and the substrate was added, followed by incubation for 1 h. All incubation steps were performed at room temperature. Finally, the absorbance of the tested samples was measured at 450 nm with the reference wavelength at 570 nm using a microplate reader. The concentration of cytokines of the positive control (only LPS) was defined as 100%. The results from the test substances were calculated as a percentage of the positive control. 

### 2.8. Statistical Analysis

Experiments were performed in triplicate and data were expressed as mean ± standard deviation (SD). Comparison of means was analyzed by one-way analysis of variance (ANOVA) using SPSS statistical software version 17.0 (SPSS Inc., Chicago, IL, USA). Differences were considered significant if *p*-value < 0.05.

## 3. Results

### 3.1. Standardization of Sample Plant Extracts by HPLC Analysis

The sample plant extracts from *C. johnstonii* were standardized by HPLC analysis. The HPLC chromatogram of the acetone fraction from *C. johnstonii* extract is shown in Figure 2. The results suggest that the major active components of *C. johnstonii* rhizomes are two flavonoid glycosides: rutin (RT) and isorhamnetin-3-o-rutinoside (IRR). The retention times of RT and IRR were about 24.5 and 26.5 min, respectively. After purification of the hydrolysis product, the samples were characterized using HPLC analysis with photodiode array, which has been widely used as an effective method for identifying unknown compounds with high sensitivity and accuracy [22]. As shown in Figure 3, the retention time of RT and IRR were about 9.0 min and 10.0 min, respectively, whereas the retention times of their hydrolysis products were about 16.0 min and 20.0 min, respectively. The hydrolysis products of RT and IRR were QT and IR, as confirmed with the similar retention time and UV spectrum of the standards. Additionally, RT and IRR were completely hydrolyzed to QT and IR after incubation with 2.0 M TFA for 30 min.

### 3.2. Determination of Antioxidant Activities of Individual Compounds

The antioxidant activities of individual flavonoid glycosides and their hydrolysis products were tested to compare their potential antioxidant activities. The results of antioxidant activities of flavonoid glycosides (RT and IRR) and their hydrolysis products (QT and IR) of *C. johnstonii* rhizomes are shown in Table 1. QT revealed the highest free radical scavenging activity, which was approximately three times higher than that of IR. Furthermore, RT had four times stronger antioxidant activity than IRR. These results show that O-methylation flavonoids (IR and IRR) from *C. johnstonii* rhizomes had lower scavenging capacity than the hydroxylation in flavonoids (QT and RT). On the other hand, the results from this study clearly exhibit that the flavonoid glycoside (RT and IRR) from *C. johnstonii* rhizomes had less antioxidant activity than their hydrolysis products (QT and IR).

### 3.3. Determination of the Antioxidant Activity of Combined Compounds

Synergistic/antagonistic interaction may influence the interaction between flavonoids. Mixtures of RT, QT, IRR, and IR were therefore studied to evaluate the combined effect on antioxidant activity. The main samples and the additional samples were chosen based on the mixture of flavonoid glycosides and their hydrolysis products. It was found that the mixture of the main sample (QT) and additional sample (IR) possessed a strong synergism in antioxidant activity, as confirmed by the lowest calculated CI value, shown in Table 2. The CI values from the mixture were less than 1, indicating a synergistic effect. Moreover, each mixture of both the main samples RT, IR, and IRR and the additional samples IRR, QT, and RT also revealed synergy. In contrast, the mixture of QT and RT or IR and IRR had a calculated CI value of more than 1, indicating antagonism. 

### 3.4. Test of Cytotoxicity against Normal Cells

The cell viability of normal cells was evaluated in order to confirm that each active substance of the *C. johnstonii* extract did not cause serious damage to healthy normal cells. Various concentrations (0–180 μM) of RT, QT, IR, and IRR were treated with PBMCs as a model of normal cells. Figure 4 shows that QT and IRR slightly decreased the viability of PBMC. An insignificant higher reduction in PBMCs viability was observed from RT and IR. However, the decrease in cell viability of all tested samples was not more than 30%. No significant difference between samples was found.

### 3.5. Cytotoxicity Study against Cancer Cells

The antiproliferative effect against cancer cells is an important mechanism of cancer prevention on the promotion and progressive stages in the cancer development process. RT, QT, IRR, and IR were therefore tested regarding their cytotoxicity against different types of cancer cells, as presented in Table 3. QT clearly stands out from the other compounds in cytotoxic activity, exhibiting strong cytotoxic activity towards various cancer cells with IC_50_ values between 6.0 ± 0.1 µM and 155.3 ± 107.4 µM. QT showed more cytotoxicity against leukemic cells as compared to cervical carcinoma cells and breast cancer cells. Furthermore, IR also showed high cytotoxic activity against EoL-1, MV4-11, K562, K562/ADR, and KB-3-1 cells with IC_50_ values between 5.3 ± 0.1 µM and 67.5 ± 23.2 µM. The flavonoid glycosides RT and IRR revealed less potential for cytotoxicity than that of their hydrolysis products: QT and IR. 

### 3.6. Cytotoxicity Study against Cancer Cells in Combination

Because of their strong cytotoxicity activity, QT and IR were chosen as combination samples to examine their potential synergism. The concentrations of the main sample were varied whereas 20% inhibition concentration (IC_20_) of additional sample (5 μM of IR and 10 μM of QT) was used for mixtures. As shown in Table 4, the CI index of the QT and 5 μM of the IR mixture revealed values less than 1 in both K562 cells and K562/ADR cells, indicating synergistic effects. Surprisingly, the mixture of IR and 10 μM of QT displayed a CI index value higher than 1 in both K562 cells and K562/ADR cells, suggesting an antagonistic effect. It is interesting that QT and IR improved the cytotoxicity effect in a synergistic manner only when QT was the main sample and IR was an additional sample.

### 3.7. Determination of Pro- and Anti-Inflammatory Activity

To support the potential of cancer prevention, anti-inflammatory activity was tested by evaluating the reduction of pro- and anti-inflammatory cytokines. The results from all tested samples are illustrated in Figure 5. The secretion of pro-inflammatory IL-6 significantly decreased in the presence of QT and IR as well as the positive control dexamethasone (DEX), whereas RT and IRR did not show any reduction effect on IL-6 secretion. The anti-inflammatory cytokine IL-10 was significantly reduced in all tested samples by at least 50%. QT and IR showed activity in decreasing IL-10 secretion that was comparable to DEX. However, the reduction of IL-10 secretion by RT and IRR was significantly less than that caused by QT and IR.

## 4. Discussion

Numerous studies have reported that flavonoids have shown potential for cancer prevention. *C. johnstonii* is enriched with flavonoids which have potential for cancer prevention. In our previous study, the acetone extract of *C. johnstonii* showed high antioxidant activity and cytotoxicity against both sensitive and resistant cancer cells. This extract promoted cancer cell cycle arrest, which occurred at the G2/M phase, followed by apoptosis. RT and IRR are the major constituents of *C. johnstonii* extract [13]. In this study, hydrolysis products (QT and IR) from *C. johnstonii* extract, especially in combination components, exerted strong proliferation inhibition of cancer cells. Both antioxidant and anti-inflammatory actions were also involved in *C. johnstonii’s* flavonoid anticancer effects. An imbalance between ROS-generating and antioxidant defense systems produced ROS accumulation. The increased generation of ROS causes DNA damage. The genetic mutations can lead to the development of cancer. Flavonoids have the potential to prevent cancer by functioning as an antioxidant in normal cells or as a pro-oxidant in cancer cells [4]. Flavonoids can scavenge ROS, chelate metal ions, and stimulate production of antioxidant enzymes [23]. Flavonoids’ health benefits are mainly derived from their antioxidant activity, which is structure-dependent [23]. The structure of flavonoids consist of the C6–C3–C6 skeleton, namely rings A, C and B. Flavonoids are generally found in glycosylated form [24]. RT and IRR are the O-glycosides and their sugar residues are rutinosides. QT and IR are aglycones without linked sugars. IR differs from QT by the presence of O-methylation in the C3’ position [25]. Our results showed that QT and IR exhibited stronger antioxidant activities than RT and IRR. The highest level of synergism in antioxidant activity was observed from the mixture of QT and IR. The existence of planar structure, the double bond in C2–C3, and hydroxy groups in positions 3 and 5 of QT and IR confer higher antioxidant activity than RT and IRR [26]. Generally, the antioxidant activity of flavonoids involves the number and location of hydroxyl groups. The most significant radical scavenging capacity is found from ring B hydroxyl configuration. Ring B hydroxyl groups donate electrons and hydrogens to free radicals, forming relatively stable flavonoids radicals [27]. The O-methylation in the C3’ position, therefore, decreases the scavenging capacity of flavonoids [28]. The O-methylation in the B-ring can suppress antioxidant activity due to the infestation of planarity, leading to electron delocalization and flavonoid phenoxyl radical [14]. It has been reported that flavonoid glycosides had weaker antioxidant activity than their corresponding aglycones [14]. The antioxidant ability of O-glycosides was lower than C- glycosides [29,30]. The glycosylation of flavonoids on OCH3 in the C-ring can reduce the free radical scavenging by the interference of the co-planarity of the B-ring, reducing the number of hydroxy groups and electron delocalization ability [31]. This provides an explanation for the finding that the hydrolysis products of flavonoid glycosides from *C. johnstonii* extract exhibit higher antioxidant activity than the flavonoid glycosides due to de-glycosylation. Although flavonoid glycosides normally present lower antioxidant capacity than aglycones, the bioavailability of aglycones is sometimes improved by glycosylation. The hydrophilicity of QT is enhanced by adding glucose moiety to at least one OH-group [32]. By the way, it was demonstrated that the mixture of the initial compound and its hydrolysis product did not provide a synergistic effect, except for the mixture of IRR as the main compound and IR as the additional sample. It might be hypothesized that this synergism is caused by the regeneration mechanism or the recycling of hydrogen from one antioxidant to another antioxidant [33]. The regeneration might occur when a lower reduction potential antioxidant sacrifices itself to protect another higher reduction potential antioxidant [34]. The mechanism of antagonism is still unclear. However, one possible explanation might be due to the interference of the glycosides at C3 positions, which could reduce their antioxidant activity [14].

Uncontrolled proliferation is one of cancer’s main characteristics. The oncogenic gene overexpression causes activation of anti-apoptotic proteins, down regulation of pro-apoptotic proteins and cellular proliferation in cancer cells [4]. Flavonoids could have an antiproliferative activity on cancer cells via induction of apoptosis, which is a programmed cell death. Many reports have shown that QT had a strong antineoplastic action. QT possessed antiproliferative activity by regulating the cell cycle, inducing apoptosis, and reducing Ras protein expression [35,36,37]. Hydroxylation was an important factor affecting flavonoids’ tumor cell growth-inhibitory effects. 3-OH, 6-OH, and 5, 7-diOH were essential structural features required for antiproliferative activity. However, it is worth noting that IR had better cytotoxicity activity than QT against EoL-1 and MV4-11 cells. Another study provided evidence to increase the potential of IR as an anticancer agent [24]. IR was shown to have lower aflatoxin B1 (AFB1)-mediated oxidative stress than QT in hepatocellular carcinoma cells [36]. IR acts as an antiproliferative agent by suppressing COX-2 protein expression, inhibiting the cell cycle protein (farnesyl protein transferase FPTase), and stimulating necrosis and apoptosis [38,39,40]. O-methylation was essential for cancer cell growth inhibitory activity. Besides, the presence of the 3′-methoxy group resulted in the improvement of biological activity [41]. In our findings, the flavonoid glycosides RT and IRR revealed less potential for cytotoxicity than their hydrolysis products: QT and IR. Our result was in line with previous studies showing that RT had less cytotoxicity than QT against many cancer cells including those found in colon, breast, hepatocellular, leukemic, and lung cancers [42,43,44]. The reason QT and IR exhibit stronger cytotoxicity than RT and IRR also lies within their chemical structures. As such, the saturation of the C2–C3 bond and the number of substituted hydroxyl groups might affect anticancer activity [45]. Besides, the steric hindrance effect from glycosidic substitutions or the sugar moiety of flavonoids at the A and/or C ring resulted in the complicated penetration through the cell membrane and the blocking of receptor binding [46,47]. Notably, both QT and IR revealed more cytotoxicity against K562/ADR than against sensitive K562 cells. These results were in line with a previous study that showed that the IC_50_ of QT for K562/ADR cells was less than half of that of K562 cells [48]. The mechanism underlying the synergistic effect of combining QT and IR is still unclear. QT and IR have similar structures and differ only in the OCH3 position on the B ring, which can be selectively incorporated into the binding pocket of the ligand, due to the polarity and the orientation of QT and IR [49]. This suggests that QT and IR might affect the ligand in a different manner and exert synergism or antagonism.

Cancer development inevitably involves chronic inflammation. Many flavonoids have been reported to exhibit anti-inflammatory properties by suppression of inflammatory cytokines and transcription factors [50]. The suppression of inflammatory-related cytokines (IL-6 and IL-10) can inhibit tumor cell proliferation and metastasis via immunosuppression [51]. The anti-inflammatory result in this study was in line with a previous study in which the high concentration of QT (100 µM) inhibited the IL-10 secretion [52]. The effect of QT in reducing or preventing inflammation has been related to the inhibition of mitogen-stimulated immunoglobulin secretion of IgG, IgM, and IgA isotypes in vitro [53]. It has been reported that IR might suppress IL-6 via a downstream target of the transcription factor, NF-κB [54]. It was shown that four hydroxylations at positions 5, 7, 3′ and 4′, together with the double bond at C2–C3 and position 2 of the B ring are likely to inhibit inflammation [54]. This study clearly exhibited that both 4′- hydroxylation and 4′-methoxylation of the B ring provided the anti-inflammatory capacity of flavonoids according to the strong reduction of IL-6 and IL-10 secretion by QT and IR. Besides, the sugar moiety of flavonoids at the C ring gave low anti-inflammatory properties. A previous study reported that the lower anti-inflammatory effect of flavonoid glycosides may be due to lower lipophilicity and steric hindrance leading to decreased membrane permeability [55]. 

## 5. Conclusions

The potential cancer prevention of flavonoid glycosides (RT and IRR) and the hydrolysis products of flavonoid glycosides (QT and IR) from *C. johnstonii* extract were evaluated in this study. These compounds were compared and investigated regarding their combined antioxidant, anti-inflammatory and antiproliferative (toward cancer cell lines) effects. The hydrolysis products of flavonoid glycosides exerted superior antioxidant, antiproliferative, and anti-inflammatory activities compared to their corresponding flavonoid glycosides. The antioxidant activity of the compounds decreased in the order QT > IR > RT > IRR. Further studies also indicated that EoL-1, MV4-11, Molt-4, U937, K562, and K562/ADR cells were sensitive to QT, while EoL-1, MV4-11, KB-3-1, K562, and K562/ADR cells were sensitive to IR. All bioactive substances showed less toxicity toward normal cells. The results of an anti-inflammatory assay also supported the promising cancer prevention effect of hydrolyzed flavonoids of *C. johnstonii.* The reduction of IL-6 and IL-10 secretion by QT and IR was higher than that induced by RT and IRR. For the combined effect on antioxidant activity, the mixture of QT and IR possessed the highest level of synergism. The mixture of the main sample (QT) and additional sample (IR) possessed a potent tumor cell growth inhibitory effect against K562 cells and K562/ADR cells. Taken together, the 3′, 4′ hydroxylation and methoxylation were important structural features required for antioxidant, antiproliferative, and anti-inflammatory effects of flavonoids products from *C. johnstonii*. The combination of the hydrolysis products of *C. johnstonii* mostly yielded synergistic effects, whereas the combination of flavonoid glycosides revealed antagonistic effects. It can be concluded that hydrolysis products from *C. johnstonii* extracts exerted potent cancer chemopreventive properties due to their promising antioxidant, antiproliferative and anti-inflammatory activities.

## Figures and Tables

**Figure 1 molecules-27-03226-f001:**
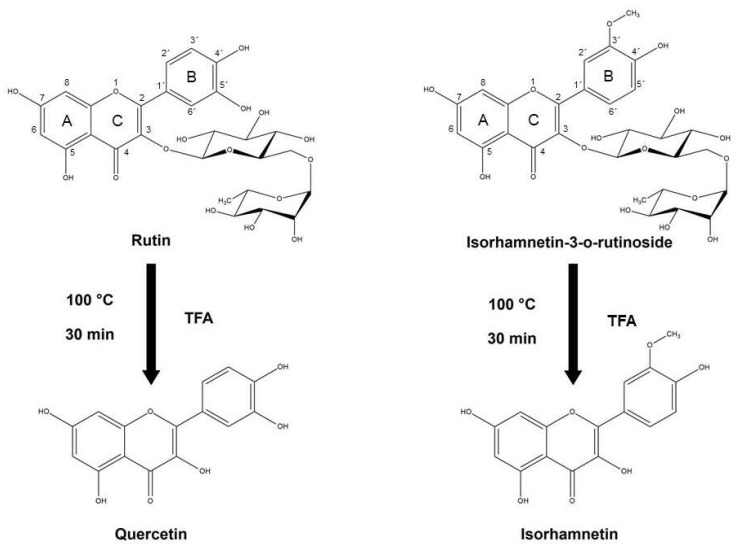
The hydrolysis scheme of flavonoid glycosides (RT and IRR) to obtain their hydrolysis products (QT and IR).

**Figure 2 molecules-27-03226-f002:**
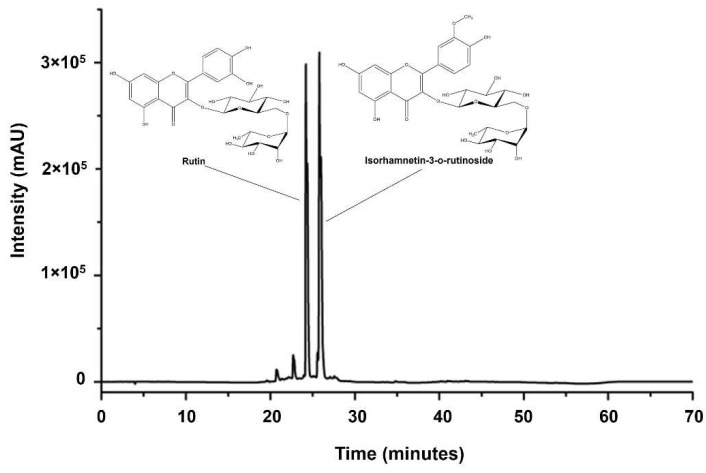
Chromatogram of RT (**left**) and IRR (**right**) from the extract of *C. johnstonii*.

**Figure 3 molecules-27-03226-f003:**
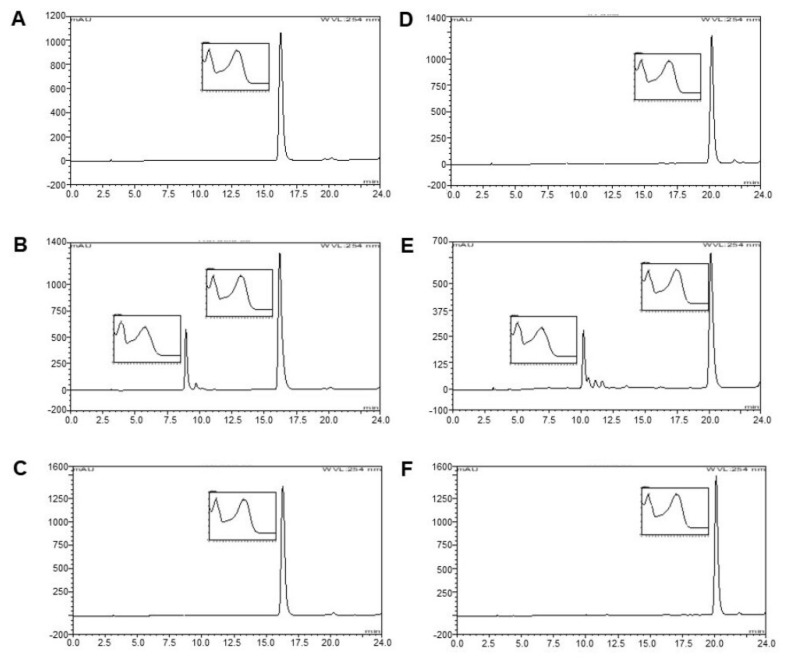
Chromatogram of standard QT (**A**), QT hydrolyzed from RT using 0.2 M TFA for 30 min (**B**), QT hydrolyzed from RT using 2.0 M TFA for 30 min (**C**), standard IR (**D**), IR hydrolyzed from IRR using 0.2 M TFA for 30 min (**E**), and IR hydrolyzed from IRR using 2.0 M TFA for 30 min (**F**).

**Figure 4 molecules-27-03226-f004:**
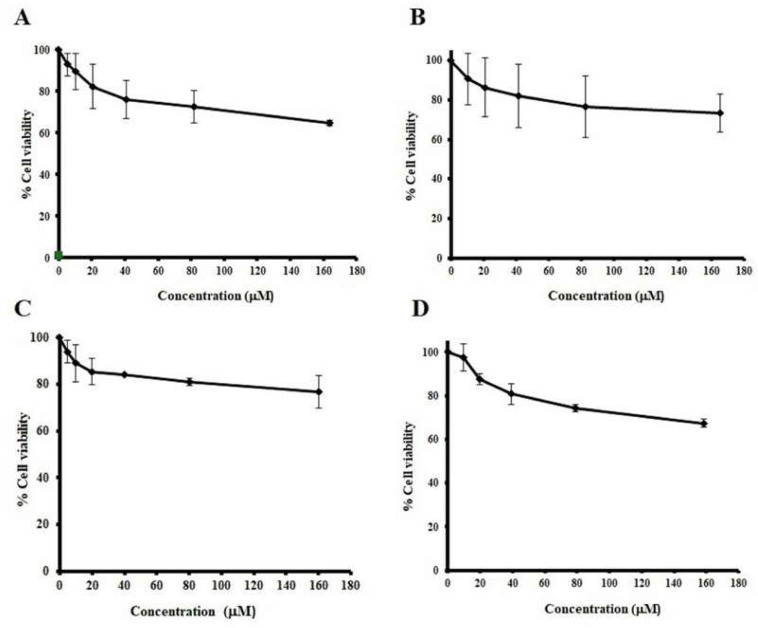
Dose-response curves between normal cells (PBMCs) viability and RT (**A**), QT (**B**), IRR (**C**), and IR (**D**).

**Figure 5 molecules-27-03226-f005:**
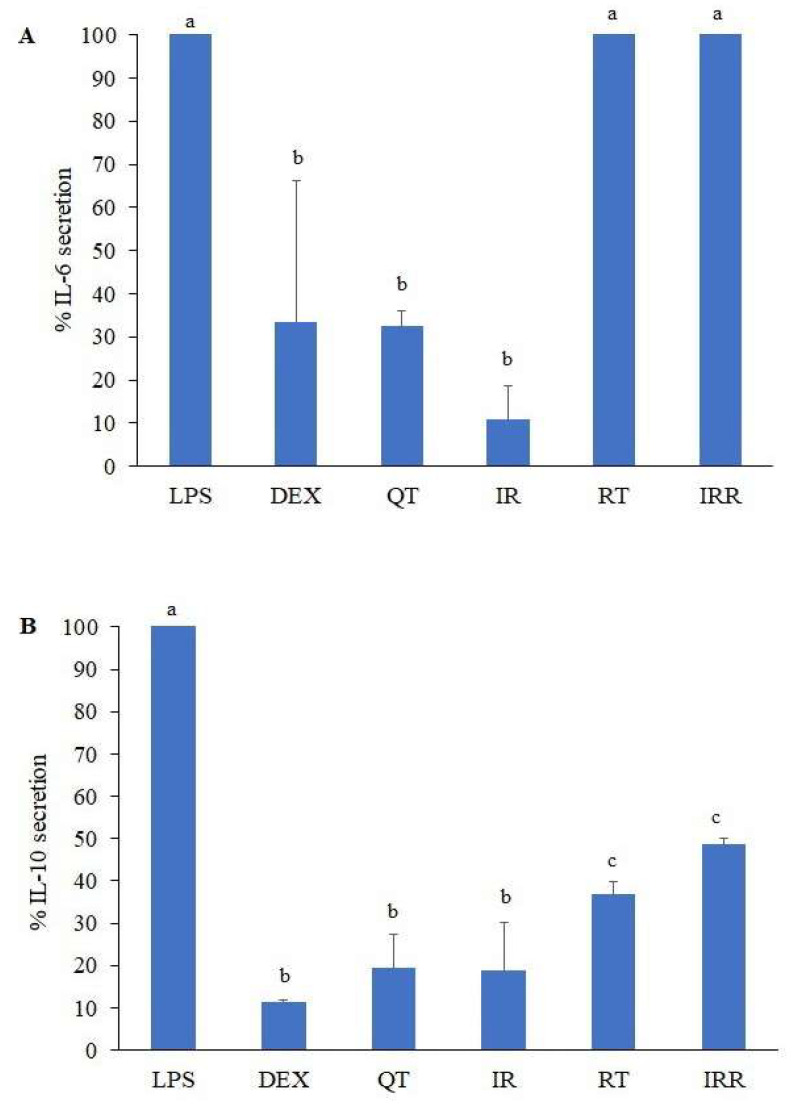
Influence of QT, RT, IR and IRR on the secretion of IL-6 (**A**) and IL-10 (**B**) as determined by ELISA assays. Different letters show statistical significance (*p*-value < 0.05).

**Table 1 molecules-27-03226-t001:** The antioxidant activity of a single compound expressed as 50% effective concentration (EC_50_) and Trolox equivalent antioxidant capacity (TEAC).

Sample	EC_50_ (µM) *	TEAC (mM) *
RT	386.0 ± 21.4 ^a^	18.8 ± 4.4 ^a^
QT	78.5 ± 4.8 ^b^	191.5 ± 15.1 ^b^
IRR	1647.0 ± 109.7 ^c^	5.94 ± 0.5 ^c^
IR	222.3± 17.0 ^d^	68.9 ± 7.3 ^d^

* Difference letters shows statistical significance (*p*-value < 0.05) in each column.

**Table 2 molecules-27-03226-t002:** The combination index (CI) on antioxidant activity in combination.

Sample	Additional Sample	CI
QT	RT	1.13
	IR	0.62
RT	QT	1.05
	IRR	0.82
IR	IRR	1.32
	QT	0.85
IRR	IR	0.88
	QT	1.56

**Table 3 molecules-27-03226-t003:** The cytotoxicity against cancer cells of a single compound expressed as 50% inhibitory concentration (IC_50_).

Cancer	Cell Type	IC_50_ (µM) *			
		RT	QT	IRR	IR
Leukemia	EoL-1	>200.0 ^a^	6.0 ± 0.1 ^b^	118.3 ± 20.6 ^c^	5.3 ± 0.1 ^d^
	MV4-11	>200.0 ^a^	20.4 ± 6.6 ^b^	>200.0 ^a^	5.9 ± 1.7 ^c^
	Molt4	>200.0 ^a^	155.3 ± 107.4 ^a^	>200.0 ^a^	>300.0 ^a^
	U937	>200.0 ^a^	28.8 ± 7.8 ^b^	>200.0 ^a^	>300.0 ^a^
	K562	>200.0 ^a^	28.7 ± 3.6 ^b^	>200.0 ^a^	45.9 ± 11.4 ^c^
	K562/ADR	>200.0 ^a^	20.9 ± 3.0 ^b^	>200.0 ^a^	25.1 ± 7.3 ^b^
Cervical carcinoma	KB-3-1	>200.0 ^a^	>300.0 ^a^	>200.0 ^a^	67.5 ± 23.2 ^b^
Breast cancer	MCF-7	>200.0 ^a^	>300.0 ^a^	>200.0 ^a^	>300.0 ^a^

* Difference letters shows statistical significance (*p*-value < 0.05) in each row.

**Table 4 molecules-27-03226-t004:** The combination index (CI) on cytotoxicity against K562 cells and K562/ADR cells in combination.

Cell Type	Sample	Additional Sample	CI
K562	QT	IR	0.68
	IR	QT	1.14
K562/ADR	QT	IR	0.92
	IR	QT	1.22

## Data Availability

Not applicable.

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
