# Peer review of "Hydrolyzed Flavonoids from Cyrtosperma johnstonii with Superior Antioxidant, Antiproliferative, and Anti-Inflammatory Potential for Cancer Prevention"

_molecules, 2022, doi:10.3390/molecules27103226_

Round 1

Reviewer 1 Report

Comments to authors:

-The current study is very interesting; however, the authors should address the following comments to improve the quality of the manuscript: 

- The manuscript should be revised for English editing and grammar mistakes.

- Please write the scientific names in the correct form all over the manuscript and in the References section (should be italic).

Title:

I think the work would benefit from the title that contains the main conclusion of the study (should be derived from the conclusion). Please modify the title.

Abstract:

- The abstract must illustrate the used methods and the most prevalent results (give more hints about methods and results). Besides, rephrase the aim of the work and the main conclusion of your findings.

-A graphical abstract is recommened.

-Introduction: (it needs to be more informative):

-Add more details about cancer (predisposing factors, causes, and mechanism of disease occurrence). Besides, illustate the prevalence of cancer in your country and all over the world if available.

- Illustrate the most common side effects of chemotherapy of Cancer.

- Explain in detail the role of ROS in the progression of various tumours.

- Illustrate the mechanism of Antioxidants and Flavonoid Glycosides in prevention of cancer progression (From previous investigations, if available).

-Rephrase the aim of the work to be clear and better sound. Add clearly the hypothesis, aims and goals of this work to the last paragraph to the introduction.

Materials and methods:

-Add Company, City, and Country of all used chemicals and reagents.

-Determination of the antioxidant activity of individual compounds: Please explain your methods in detail.

-Cytotoxicity study against cancer cells: Support your methods with specific references.

- Cytotoxicity against normal cells: Support your methods with specific references.

-2.7.2. ELISA study:

• Please discuss in detail. Besides, specific references should be added

-Statistical analysis:

-Add more details about the software used in the statistical analyses.

-Results: Good presentation

- Please add a starting paragraph to the results section to briefly introduce the topic, your goals and hypothesis and a short summary of what you did in this work.

-Please, increase the resolution of all figures (should be 600 dpi). Besides, colored figures are recommended.

-Discussion:

- The authors are advised to illustrate the real impact of their findings without repetition of results.

-Try to Illustrate the role and mechanism of Flavonoid Glycosides and Their Hydrolyzed Products in prevention of cancer progression.

-Conclusion

- Should be rephrased to be sounded. A real conclusion should focus on the question or claim you articulated in your study, which resolution has been the main objective of your paper?

Author Response

Dear Reviewer,

We would like to thank you for the opportunity to revise and improve our manuscript. We appreciate all the suggestions and comments. All changes in the manuscript are marked up using the “Track Changes” function. Our manuscript follows the suggested and each comment is answered as attached file.

We, therefore, believe that the revision in our manuscript would be satisfactory to publish in the Molecules Journal. We confirm that neither the manuscript nor any parts of its content are currently under consideration or published in another journal. All authors have approved the manuscript and agree with its submission to Molecules.

Sincerely Yours

Ruttiros  Khonkarn

(Corresponding author)

Reviewer 2 Report

The authors describe an interesting study about flavonoid glycosides (RT and IRR) and the hydrolysis products of flavonoid glycosides (QT and IR). Scientifically, I have no criticism. However, following points needs to be addressed prior to the consideration of the acceptance of the paper

  1. It is suggested to perform NMR analysis of hydrolysis products of flavonoid glycosides prepared by chemical hydrolysis using trifluoroacetic acid and discuss about it.

  1. It is recommended to cite following relevant article along with ref:17 for the importance of C-glycosides

Sateesh Dubbu, Ande Chennaiah, Ashish Kumar Verma, Yashwant D. Vankar, Stereoselective synthesis of 2-deoxy-β-C-aryl/alkyl glycosides using Prins cyclization: Application in the synthesis of C-disaccharides and differently protected C-aryl glycosides, Carbohydr. Res. 2018, 468, 64–68

Author Response

(The authors gave the same response as above.)

Round 2

Reviewer 1 Report

The authors have carried out significant changes to the manuscript. They have addressed all the suggested corrections and comments. Really, it's an interesting study that has a significant impact. Now, the manuscript could be accepted.

Congratulations.